# The Fabrication of Docetaxel-Containing Emulsion for Drug Release Kinetics and Lipid Peroxidation

**DOI:** 10.3390/pharmaceutics14101993

**Published:** 2022-09-21

**Authors:** Yifang Wu, Mengmeng Wang, Yufan Li, Hongmei Xia, Yongfeng Cheng, Chang Liu, Ying Xia, Yu Wang, Yan Yue, Xiaoman Cheng, Zili Xie

**Affiliations:** 1College of Pharmacy, Anhui University of Chinese Medicine, Hefei 230012, China; 2Clinical College of Anhui Medical University, Hefei 230601, China; 3School of Life Science, University of Science and Technology of China, Hefei 230027, China; 4Anhui Institute for Food and Drug Control, Hefei 230051, China

**Keywords:** docetaxel, emulsion, drug release, lipid peroxidation

## Abstract

Docetaxel (DTX)-based formulation development is still confronted with significant challenges, due to its refractory solubility and side effects on normal tissues. Inspired by the application of the transdermal drug delivery model to topical treatment, we developed a biocompatible and slow-release DTX-containing emulsion via self-assembly prepared by a high-speed electric stirring method and optimized the formulation. The results of accelerated the emulsion stability experiment showed that the emulsion prepared at 10,000 rpm/min had a stability of 89.15 ± 2.05%. The ADME, skin irritation, skin toxicity and molecular interaction between DTX and excipients were predicted via Discovery Studio 2016 software. In addition, DTX addition in oil or water phases of the emulsion showed different release rates in vitro and ex vivo. The DTX release ex vivo of the DTX/O-containing emulsion and the DTX/W-containing emulsion were 45.07 ± 5.41% and 96.48 ± 4.54%, respectively. In vitro antioxidant assays and anti-lipid peroxidation models revealed the antioxidant potential of DTX. However, DTX-containing emulsions could maintain and even enhance the antioxidant effect, both scavenging free radicals in vitro and inhibiting the process of lipid peroxidation.

## 1. Introduction

Docetaxel (DTX) is an antitumor drug [1,2] and plays an important role in the treatment of breast cancer [3] and prostate cancer [4], among others. However, it has different degrees of similar side effects to some chemotherapeutic drugs, such as pericardial effusion [5], fluid retention [6] and radiation recall dermatitis (RRD) [7], which all can affect the treatment outcome. Additionally, due to its bulky polycyclic structure, the insolubility of DTX (4.93 μg/mL in water) becomes a disadvantage [8]. Therefore, to enhance DTX solubility and to achieve sufficient DTX concentrations for clinical applications, researchers over the last few years have been committed to new dosage forms, such as nanoparticles [9,10,11], liposomes [12], microspheres [13], micelle [14,15] and lipid emulsions [16,17].

Paclitaxel microspheres modified by tyrosine were investigated a decade ago, and it was reported that they could penetrate the epidermal layer via local skin administration [18]. In addition, previous research has also proved that paclitaxel ointment can be used locally to treat psoriasis [19]. Based on the feasibility of transdermal administration, we chose the most common external dosage form of emulsion. Emulsions belong to the transdermal drug delivery system (TDDS), where first-pass effects can be avoided by topical dermal administration. Recently, particle systems have been widely exploited to increase the solubility of insoluble drugs also with antitumor activity and to reduce their toxicity [20,21]. DTX encapsulated inside the micelle structure of the emulsion can reduce direct skin irritation [22]. Taking into account the possibility of skin inflammation during DTX treatment [23], we combined computer software to predict skin irritation and the toxicity of each chemical component in the emulsion. Computer-aided drug research has been involved in target docking, toxicological prediction, pharmacophore analysis, etc. [24,25].

Recently, more and more attention has been paid to the relationship between the destruction of lipid peroxidation and diseases. Lipid peroxidation is mainly divided into two reactions: one is an enzyme-mediated reaction, such as lipoxygenase (LOX), cyclooxygenase (COX), cytochrome P450, etc., and the other is a free radical chain reaction initiated by transition metals such as Fe and Cu [26]. Polyunsaturated fatty acids (PUFA) are the substrate of lipid peroxidation and the main component of lipids. They can easily provide hydrogen atoms through carbon–carbon double bonds. As the main component of cell membranes, lipids play an important role in maintaining the integrity of the cell structure. Once lipid peroxidation occurs, lipids are the first to participate in the reaction. During the reaction, a large number of reactive oxygen (ROS) and some oxidation products are produced, which affects the morphology and function of cells and increases the possibility of inflammation [27]. In addition, long-term inflammation causes the cells to remain at the level of oxidative stress. Oxidative stress is one of the main factors of membrane damage and also one of the reasons leading to the malignant development of diseases such as cancer (Figure 1). The formation of malondialdehyde (MDA) represents the state of oxidative stress and is also an indicator of lipid peroxidation. In a large number of literature reports, antioxidants can effectively inhibit lipid peroxidation and prevent mild inflammation from moving to malignant tumors [28]. Paclitaxel (PTX) has been reported to have a certain antioxidant potential [29]. PTX and DTX belong to taxane. From the perspective of chemical structure, based on the mother nucleus of taxane, the substituents at the C10 position of the two are different, and PTX is acetyl, whereas DTX is hydroxyl. However, there are few reports of the antioxidant activity of DTX.

Therefore, the objective of this study was to develop a safe and stable DTX-containing emulsion, to investigate the antioxidant potential of DTX and DTX-containing emulsion and to establish two different models of lipid peroxidation reaction ex vivo to explore the relationship between DTX and lipid peroxidation.

## 2. Materials and Methods

### 2.1. Materials and Reagents

DTX (purity 98.0%) was purchased from Chengdu Pufei De Biotech Co., Ltd. (Chengdu, China). Glycerol, Stearic acid, 1,1-diphenyl-2-picric acid hydrazine (DPPH) and Thiobarbituric acid (TBA) were provided by Shanghai Yuanye Biology Science and Technology Co., Ltd. (Shanghai, China). Vaseline was provided by Shandong LIRCON Medical Technology Co., Ltd. (Dezhou City, Shandong, China). Sodium dodecyl sulfate (SDS) was provided by Tianjin Guangfu Chemical Research Institute (Tianjin, China). Hydrogen peroxide(H_2_O_2_) was provided by Shanghai SuYi Chemical Reagent Co., Ltd. (Shanghai, China). Trichloroacetic acid (TCA) was provided by DAMAO Chemical Reagent Factory (Tianjin, China). All the other chemicals and solvents were of analytical reagent grade.

### 2.2. Animals

Healthy Kunming mice (male, 20 ± 2 g) were purchased from the Animal Experimental Center of Anhui University of Chinese Medicine (Hefei, China). All animal experiments complied with the guidelines approved by the ethics committee of Anhui University of Chinese Medicine (Hefei, China). The animals were raised under constant environmental conditions (25 ± 2 °C, 40–70% relative humidity). The animals were free to access food and sterile water.

### 2.3. Preparation and Optimization of DTX-Containing Emulsion

#### 2.3.1. Selection of the Ratio of Oil Phase Composition

Vaseline is chemically inert and is often used as the auxiliary material of emulsion, which plays a lubricating function. Stearic acid, as an amphiphilic molecule, plays an important role in emulsification. In addition, we added flaxseed oil to the emulsion, because it contains a lot of α-linoleic acids (ALA) that can increase affinity and permeability to the skin. In the experiments, the two kinds of dosage ratios of stearic acid and Vaseline (1:1 and 2:3), and whether prescriptions needed flaxseed oil added, were investigated, as described in Table 1.

#### 2.3.2. Preparation of DTX-Containing Emulsion

DTX-containing emulsions were prepared using a high-speed electric stirring method. Hydrophilic and lipophilic components were separately weighed according to the prescription and were placed in two 50 mL beakers, respectively. Then, they were melted in a water bath at 75 °C. After complete dissolution, the water phase was added to the oil phase, and it was then emulsified for 5 min at shear speeds of 500 rpm/min, 5000 rpm/min and 10,000 rpm/min. The emulsion samples were placed on a slide and were spread evenly with a cover slip, and then the droplet distribution and morphology of the emulsion were observed using the optical microscope according to the method of Lee et al. [30]. An amount of 1 mL DTX with a concentration of 5 mg/mL was added to the oil phase or water phase to prepare the emulsions, namely DTX/O and DTX/W. In addition, the blank emulsion without DTX was prepared and defined as a blank emulsion (B).

#### 2.3.3. Determination of Drug Recovery

The drug recovery (DR) of DTX-containing emulsion was measured to check the accuracy of the drug content determination method. Briefly, 1.8 mL of absolute ethanol was added to 200 μL of DTX-containing emulsion by sonicating for 5 min and 10 min to destroy the micelle structure of the emulsion. The drug content in the sample solution was determined at 230 nm by ultraviolet spectrophotometry (1600 UV-Vis, Shanghai Mepeda instrument Co., Ltd., Shanghai, China). The DR of the DTX-containing emulsion was calculated, as below:(1)$$DR=\frac{Amount\;of\;DTX\;contained}{Amount\;of\;DTX\;added}\times 100\%$$

#### 2.3.4. Accelerated Emulsion Stability

The accelerated emulsion stability was referred to with the method described in [30], with some modification. A fresh emulsion (6 mL) was placed into a 10 mL centrifuge tube and was centrifuged at 7500 rpm/min for 15 min at 25 °C. Centrifugation caused the unstable emulsion to divide into three layers, which were the oil layer (top), emulsion layer (middle) and aqueous layer (bottom). The initial height of the emulsion (*H_0_*) and the height of the emulsion layer (*H_e_*) were measured. The emulsion stability (*ES*) was calculated using the equation below.
(2)$$ES=\frac{H_{e}}{H_{0}}\times 100\%$$

### 2.4. Drug Release across the Dialysis Membrane Experiment In Vitro

A small stirring magnet was placed in the Franz diffusion cell and was filled with PBS solution (pH = 7.4). The dialysis membrane (MWCO: 8000–14,000) was soaked in boiling water and was then taken out and placed between the upper and lower compartments of the diffusion bottle in order to make it fully contact the PBS solution. An amount of 1.5 mL DTX solution, blank emulsion, DTX/W-containing emulsion and DTX/O-containing emulsion as samples were put into the upper compartment. The diffusion experiment was carried out at a constant temperature of 37 ± 1 °C, controlled by a constant-temperature magnetic agitator. An amount of 2 mL of solution from the lower compartment of the chamber was collected at different times (0.5, 1, 2, 3, 4, 5, 6, 7, 8, 9, 10, 11, 12, 24, 36, 48, 60, 72, 84 and 96 h), and a PBS buffer with the same volume and temperature was added. Thereafter, their absorbance was measured at 230 nm and recorded. The test was repeated three times. The percentage of the released DTX at each time point was calculated using the following equation:(3)$$Released\;DTX\;\left(\%\right)=\frac{Amount\;of\;DTX\;at\;time\;}{Total\;amount\;of\;DTX}\times 100\%$$

### 2.5. Molecules Interaction Study

Molecular docking is one of the most common methods used to predict the binding conformation of a ligand with a suitable target protein. In the present study, molecular docking was performed using the software of Discovery Studio 2016. The structures of tubulin were obtained from the Protein Data Bank (PDB ID = 1 TUB). First, we looked for suitable binding sites on tubulin dimmers and performed the docking studies with the flexible ligand (DTX) and the rigid receptor (β-tubulin). The local docking was performed with a radius of 13 of a small sphere. The interaction forces between individual molecules were analyzed with the help of computer software. According to the results of the analysis, the molecular distribution in the emulsion was reasonably speculated.

### 2.6. ADME Analysis

The ADME (absorption, distribution, metabolism and excretion) characteristics of six selected compounds (DTX, stearic acid, Vaseline, flaxseed oil, SDS and glycerol) in the prescription were studied via the ADMET protocol in the Discovery Studio 2016 software package, to estimate the bioavailability of the compounds [31]. Some parameters were calculated to include atom-based Log P98 (ALogP98), ADME 2D fast polar surface area (ADME 2D FPSA), blood–brain barrier (BBB) and cytochrome P4502D6 (CYP2D6).

### 2.7. Toxicity Study

TOPKAT compound toxicological properties were used to predict the skin toxicity and skin sensitivity of the components in the prescription [32,33].

### 2.8. Antioxidant Activity Studies

#### 2.8.1. The Activity of Scavenging the DPPH Free Radical

The DPPH solution with a deep violet color has a characteristic absorption wavelength of 517 nm. The color changes into pale yellow with a value of absorbance decreasing when a non-radical form (DPPH-H) is produced. The activity of scavenging the radical DPPH· by DTX and DTX-containing emulsions can be measured at 517 nm. The method previously described was used with slight modifications to assess the scavenging free radical DPPH· by DTX and its emulsions [34,35]. Briefly, we mixed 1.0 mL of different concentrations of DTX solutions (concentration range of 10–50 μg/mL) with 2.0 mL of DPPH (0.2 mmol/L), and they reacted in the dark. The DPPH· was measured at 517 nm. The equation of the scavenging rate to the DPPH· is shown as follows:(4)$$\mathrm{Scavenging}\;\mathrm{rate}\;\mathrm{to}\;\mathrm{DPPH}\;(\%)=\left(1-\frac{A_{s-A_{c}}}{A_{0}}\right)\times 100\%$$
where “*A_0_*” is the absorbance of the blank group; “*A_S_*” is the absorbance of the sample group; and “*A_C_*” is the absorbance of the sample control group.

#### 2.8.2. The Activity of Scavenging H_2_O_2_

The solution of hydrogen peroxide (40 mmol/L) was prepared in distilled water. In the sample group, 0.6 mL DTX solution (concentration range of 100–500 μg/mL) and DTX-containing emulsions were added into EP tubes, mixed with 1.8 mL H_2_O_2_ and reacted for 10 min at 25 °C, and then they were determined at 230 nm. Simultaneously, the absorbance of the blank group with 40% ethanol and the sample control group without hydrogen peroxide solution were determined. The scavenging rate to H_2_O_2_ was calculated as shown in Equation (4).

### 2.9. Drug Release Study Ex Vivo

After being weighed, the mice were anesthetized with ethyl carbamate. The villi were removed from the backs of the mice with a shaving knife and depilating cream. Back skin with dimensions of 2 × 2 cm was cut off and washed with 0.9% normal saline. According to a previously published paper, the transdermal model and the mouse subcutaneous mucous membrane model have been established [36,37]. The skin section (complete mouse back skin or mouse subcutaneous mucosa) was mounted between the donor and the acceptance cavity of the diffusion cell, as shown Figure 2. The release kinetics studies from DTX and DTX-containing emulsions ex vivo were performed using the subcutaneous mucosa of mice at 37 ± 1 °C. A PBS buffer solution (pH = 7.4) containing 10% ethanol was added to the acceptance chamber. The remaining experimental steps were the same as in Section 2.4. Various mathematical models (zero order, first order, Higuchi, Hixson–Crowell and Koresmeyer–Peppas were used to determine the drug release kinetics and mechanism of DTX-containing emulsion, as reported prior [38].

### 2.10. Lipid Peroxidation Model of Tissue Homogenate Ex Vivo

After the back skins of the above mice were taken, the liver and spleen tissues of the mice were promptly removed. The tissues were carefully washed in 0.9% normal saline to remove blood. After weighing, the tissues added to the 0.9% normal saline (divided three times) were homogeneous, and then the 10% tissue homogenates were obtained [39].

In one model, after mixing 1.0 mL of liver tissue homogenate with 100 μL of DTX solution or DTX-containing emulsions for 5 min, 100 μL Fe^2+^ solution was added to establish a model of a normal cell’s lipid peroxidation reaction. In another model, 1 mL of spleen tissue homogenate was taken out, and 100 μL Fe^2+^ solution (10 mmol/L) was added to induce for 30 min (25 °C). Then, 100 μL of DTX solution or DTX-containing emulsions was added to the above mixture solution to build the model of the Fe^2+^-induced lipid peroxidation reaction in activated cells. The experimental steps of the two models were consistent. The tissue homogenates were mixed with 100 μL 0.9% normal saline and 100 μL Fe^2+^ as a blank control and positive control. After incubation for 1.5 h at 37 °C, the above solution was mixed with 3.0 mL TBA working liquid and was heated at 95 °C for 40 min. After cooling and centrifuging at 4000 rpm/min for 8 min, the absorption of the clarified supernatant solution was determined at a wavelength of 532 nm.

### 2.11. Statistical Analysis

The obtained results were expressed as mean ± standard deviation (SD). The statistical analyses were performed using SPSS Software 23.0 (IBM, Armonk, NY, USA) by an analysis of variance (ANOVA) with Duncan’s test. *p* < 0.05 was considered a statistically significant difference.

## 3. Results and Discussion

### 3.1. Formulation Studies

#### 3.1.1. Effects of Shear Speed on Emulsion Micelle and Stability

The micelle size and stability of the emulsion are important indicators of the quality of the emulsion. To select the appropriate emulsifying speed, the blank emulsions were prepared to emulsify for 5 min at low, medium and high shear speeds according to the same prescription (Figure 3A). The emulsion mixed at 500 rpm appeared in three separate layers after centrifugation, which may have been caused by the state of the emulsion being extremely unstable (Figure 3B). A significantly higher accelerated emulsion stability (89.15 ± 2.05%) was obtained at a shear speed of 10,000 rpm (*p* < 0.05). However, there is no difference between the shear speed of 5000 rpm and 10,000 rpm. Increasing mechanical energy reduces interfacial tension and forms a stable emulsion quickly [40]. The microphotographs of emulsions showed that the micelle dispersions of the emulsions prepared at low and medium speed were uneven, except for the micelle of the emulsion prepared at a high shear speed (Figure 3C–E). The particle size influenced the stability of the emulsion. Therefore, a shear speed of 10,000 rpm/min was selected for preparing the emulsion.

#### 3.1.2. Effects of Different Ratios of the Oil Phase on Drug Release In Vitro

The release of DTX solution was only more than 20%, which may be due to the fact that DTX solution does not create a leaky tank condition with the receiving solution in the below compartment, thus making the release of DTX difficult to detect (Figure 4). In addition, when the dosage of stearic acid to Vaseline was 2.5 g and 2.5 g (1:1) in a 10 mL system, only a small amount of DTX was released from DTX-containing emulsions in vitro. Although DTX/O (1:1) and DTX/W (1:1) suddenly released at 60 h, especially DTX/W (1:1), the in vitro total release of DTX from DTX/W (1:1) did not exceed 20% in 96 h, and DTX/O (1:1) was only half of DTX/W (1:1). However, after changing the ratio of stearic acid and Vaseline to 2:3 and adding flaxseed oil, the release of DTX from both DTX/W (2:3) and DTX/O (2:3) reached 20.82 ± 1.67 and 14.37 ± 1.73% within 12 h. Then, in the next 72 h, an additional 5~8.5% of DTX was sustained and slowly released, which could be attributed to the dissolution of DTX existing on the surface of the emulsion [11]. In addition, these results are in accordance with the study by Zhang et al. [17], who reported that DTX was lipid friendly and could easily be encapsulated in the oil phase, making it difficult to “detach”. On the other hand, DTX was freely distributed in the water phase and was weakly limited by oil phase components [14].

In combination with the above results, we completed the formulation study of DTX-containing emulsion, briefly described as follows: According to the formulation DTX (2:3) in Table 1, DTX-containing emulsion was prepared at 10,000 rpm/min after complete dissolution at 75 °C.

### 3.2. Drug Recovery

The drug recovery (DR) was calculated by determining the drug content. The drug recovery of DTX/O and DTX/W calculated by ultrasonic treatment for 5 min was 86.74 ± 2.18% and 70.21 ± 2.03%, respectively. However, with the ultrasonic time prolonging, the DR of DTX/O and DTX/W reached 99.74 ± 1.09% and 99.68 ± 1.17%, respectively, which could indicate the accuracy of the detection method.

### 3.3. Molecular Interaction Analysis between DTX and Excipients

It has been proved theoretically that docetaxel has good docking performance with β-tubulin [41]. Therefore, we also conducted a docking model of β-tubulin with DTX (Figure 5A). DTX kept a conventional hydrogen bond with amino acid residue Arg278 of β-tubulin, and hydrophobic interactions were generated with Val23, Pro360, and Leu371. In addition, it formed a mainly unfavorable bump with Arg369, His229, and Asp226. Due to different in vitro release studies of DTX/O and DTX/W, the interaction forces between the prescription components were analyzed. DTX could form hydrogen bonds with SDS and glycerol. SDS could form ionic bonds with stearic acid (Figure 5B). The interaction forces between the remaining molecules were mainly Van der Waals forces. These also showed that the structure of emulsion was stable, and there was the possibility of a slow release during the in vitro release process. As a result, we speculated on the chemical structure distribution of DTX in the oil-in-water-type emulsions, as described in Figure 5C.

### 3.4. ADME Analysis and Skin Toxicity Study

The pharmacokinetic properties of the components in formulation were investigated using Discovery Studio 2016 ADME protocol. The results of the ADME results are listed in Table 2 and Figure 6. Vaseline, flaxseed oil and SDS showed very high blood–brain barrier penetration. Except for docetaxel, the remaining five components demonstrated good or moderate levels of absorption. All the analyzing components were anticipated to be CYP2D6 non-inhibitors. Finally, the components could bind the plasma protein at a rate greater than 90%, except for glycerol, which was predicted to bind at a rate less than 90%.

Furthermore, considering that the dosage form of this study was mainly used for local skin applications, the skin toxicity of all ingredients in the prescription was evaluated. Except for Vaseline, the remaining components were not sensitive to skin. Vaseline can be used as a carrier to increase the permeability of active ingredients and can be used in combination with other active components as a dressing to promote wound healing [42]. In addition, only stearic acid and flaxseed oil in the prescription had moderate irritation to the skin, whereas the rest of the ingredients were mild or had no irritancy (Table 3). In general, the formulation of the emulsion could promote the penetration of DTX into the skin and could be relatively friendly to the skin.

### 3.5. Ex Vivo Release Study

Mouse skin and its subcutaneous mucosa were used to simulate the local release of drugs under a physiological environment. In the ex vivo drug release device, a PBS solution containing 10% ethanol was selected as the diffusion medium of the lower compartment to form a sinking condition. This was consistent with the purpose of adding Tween-80 to the diffusion medium, as reported [11]. The previous literature has also reported adding different concentrations of glutathione (GSH) to the diffusion medium to simulate the tumor microenvironment and induce DTX release [43].

In the transdermal release experiment, DTX release from the DTX solution was 15.31 ± 2.79%. Only a small amount of docetaxel was detected in DTX/O and DTX/W within 96 h, which was 3.97 ± 0.34% and 5.64 ± 0.22%, respectively (Figure 7A). Then, using sonication to handle experimental skin, the amount of docetaxel retained in the mouse skin was 72.57 ± 6.02% for DTX/O-containing emulsion and 61.96 ± 5.11% for DTX/W-containing emulsion, which indicated that the emulsion micelle structure made DTX slowly release. In addition, the retention of inflammatory skin was higher than that of normal skin, as reported by Yin et al. [19], who also showed that the inflammatory environment could improve the permeation and induce DTX enrichment.

In addition to the skin, the subcutaneous mucosa, such as ocular mucosa [44], nasal mucosa [45] and oral cavity mucosa [46] is the ideal route of administration. The drugs across the mucosa of the ocular, nasal and oral cavity can reach the target site or directly enter systemic circulation. The percentage of cumulative DTX released from the DTX solution, DTX/O and DTX/W through subcutaneous mucosa ex vivo for 96 h reached 85.73 ± 7.63%, 45.07 ± 5.41% and 96.48 ± 4.54%, respectively (Figure 7B). As is clear, more DTX molecules were released from DTX/W than that from DTX/O, which was consistent with the in vitro release of DTX-containing emulsion. Moreover, DTX-containing emulsion penetrated into the mucosa, laying the foundation for the future research of mucosal inflammation.

Using various mathematical models, the drug release curves were analyzed via Origin Software 2019b (OriginLab Corp., Northampton, MA, USA). Zero-order, first-order, Higuchi, Hixson–Crowell and Koresmeyer–Peppas are commonly used models used to describe drug release kinetics [47,48]. The correlation coefficient values (R) of each mathematical model for DTX, DTX/O and DTX/W are summarized in Table 4. For DTX solution and DTX/O, the first-order models of drug release had a higher R value, whereas for DTX/W, the R value of the zero-order model was 0.9922. This suggests that the drug release mechanism of DTX/O is more suitable for the first-order kinetic study, and the drug release mechanism of DTX/W is more suitable for the zero-order kinetic study.

### 3.6. In Vitro Antioxidant Activity Assays

#### 3.6.1. In Vitro DPPH and H_2_O_2_ Scavenging Assays for DTX

DPPH and H_2_O_2_ scavenging assays are the most effective, convenient and accurate methods for the evaluation of the antioxidant activity of chemical compounds [49]. Therefore, these two assays were used to investigate whether DTX has antioxidant activity. Within the range of 10~50 μg/mL, the DTX solution showed a logarithmic relationship with the DPPH free radical clearance rate at 4 h (Figure 8A). Additionally, within the range of 100~500 μg/mL, the DTX solution showed a linear relationship with the H_2_O_2_ clearance rate (Figure 8B). According to the relationship described above, the IC_50_ (half maximal inhibitory concentration) of DTX scavenging on DPPH free radical and H_2_O_2_ were 40.65 ± 6.29 μg/mL and 327.47 ± 52.23 μg/mL, indicating that DTX had excellent antioxidant activity. The IC_50_ of PTX to DPPH free radical and H_2_O_2_ was 1.49 mg/mL and 2.17 mg/mL [50]. Compared to the IC_50_ of DTX with that of PTX, it can be concluded that DTX is superior to PTX in terms of antioxidant activity.

#### 3.6.2. Comparison of Antioxidant Activity of DTX and DTX-Containing Emulsion

To further determine the antioxidant activity of DTX-containing emulsion and whether the excipients affect antioxidation function, the free radical removal rates among DTX solution, DTX/W and DTX/O were compared. The scavenging rate of the excipients on DPPH free radicals was 57.07 ± 6.34%, which was not significantly different from that of free DTX. However, the scavenging rate of DTX/W and DTX/O on DPPH free radicals was significantly higher than that of blank emulsion (Figure 8C). Interestingly, blank emulsion, DTX/O and DTX/W had extremely high clearance rates on hydroxyl radicals, which were 94.74 ± 6.12%, 96.54 ± 7.14%, and 98.29 ± 5.50%, respectively, showing that the excipients themselves played a greater role to help DTX clear free radicals (Figure 8D). For example, the total unsaturated fatty acids in flaxseed oil accounted for 74%, of which α-linoleic acid accounted for 57%. This unsaturated fatty acid not only provides lipid content and antioxidant function, but also has excellent permeability and affinity with the skin [51]. It is also possible that the excipients firstly exert the ability to scavenge free radicals in the long-term action and perform the antioxidant effect with the gradual release of the drugs. In the current report, H_2_O_2_ is recognized as the primary ROS-signaling molecule [52]. DTX and its emulsions all had strong scavenging stability against H_2_O_2_, which also meant that DTX could clear ROS. However, in cancer cells, ROS is 100-fold higher than that in normal cells, and the entire environment belongs to an oxidized condition. DTX induces the apoptosis of cancer cells by promoting the level of ROS. Therefore, from the present results, DTX with a low concentration sharply inhibits hydrogen peroxide in the early mild inflammation stages.

### 3.7. Lipid Peroxidation Study of DTX and DTX-Containing Emulsion in Different Tissue Homogenates

Cells ingesting too many ferrous ions triggers the Fenton reaction to initiate lipid peroxidation, which is one of the mechanisms by which ferroptosis occurs [53,54]. This characteristic of ferrous ions (Fe^2+^) was used to establish a model of lipid peroxidation, and the production of malondialdehyde (MDA), an oxidation product, was monitored. As seen in Table 5, in liver homogenate, the inhibition rate of DTX on MDA production increased slightly with increasing the concentration. There was a significant difference in the inhibition rate between 100 μg/mL DTX and 200 μg/mL DTX on MDA production. This showed that DTX to lipid peroxidation in liver tissue was insensitive when the concentration of DTX solution was between 200 μg/mL and 1000 μg/mL. This situation could also be that the metabolism of DTX by hepatic cytochrome CY4P3 enzyme attenuates the effect of DTX on the inhibition of MDA production [55].

However, the overall results showed that the production of MDA was affected by DTX, especially in spleen tissue. It has been reported that paclitaxel inhibited the hyper-activation of splenic cells by LPS [56]. Moreover, DTX is considered an immune adjuvant in vaccines [57]. In the context of an immune response, DTX has been studied and seen as a novel chemical immunomodulator that enhances CD4 and CD8 T-cell function in the spleen in the tumor environment [58]. A total of 43.26~57.02% of the inhibition rate of MDA production was obtained in the spleen homogenate. This could be attributed to the time difference in the addition of Fe^2+^ in two models. However, there was still a small disparity in the inhibition rate, which may have been caused by DTX promoting oxidation. The previous literature reports that anticancer drugs cause oxidative stress inside the cells [59,60]. Moreover, Ray et al. used DTX as an inducer to establish a lipid peroxidation model of goat livers [61].

DTX/W-containing emulsion with a high DPPH free radical scavenging rate in vitro can participate in lipid peroxidation reactions. Interestingly, compared with 500 μg/mL DTX solution, DTX/W had the same effect on inhibiting MDA production in different tissue homogenates. This demonstrates that excipients can reduce the possibility of oxidative stress induced by DTX. The emulsion can wrap DTX in the center of micelles, release DTX dosage stably and slowly, inhibit lipid peroxidation and control the oxidative stress it may produce. In addition, flaxseed oil in excipients is an excellent antioxidant [62], which can resist external oxidation reactions and play a double guarantee.

## 4. Conclusions

In summary, we successfully screened out a suitable process to prepare DTX-containing emulsion by adjusting the proportion of oil phase components and screening the sheering speed. There were mainly Van der Waals forces between each component in the emulsion, and a small number of hydrogen bonds improved the stability and permeability of the emulsion. The drug release ex vivo experiments showed that docetaxel could be retained in the skin or could directly infiltrate into the general circulation through mucosal administration for systemic treatment. Furthermore, free radical scavenging experiments in vitro showed that both DTX and DTX-containing emulsion had antioxidant capacity, and there was a synergistic effect between them. Although increasing the concentration of DTX did not regularly control lipid peroxidation, DTX affected the production of oxidation products as a whole. Therefore, taking emulsion as the platform for DTX delivery, the targeted treatment of some refractory chronic diseases makes it more meaningful in clinical practice.

## Figures and Tables

**Figure 1 pharmaceutics-14-01993-f001:**
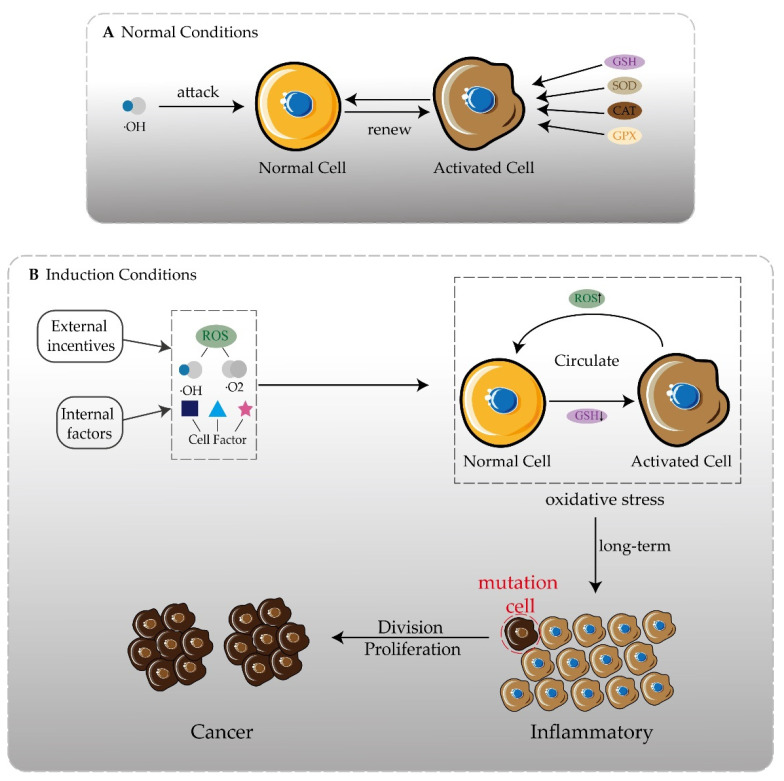
One of the inducements of disease development is oxidation. (**A**) Under normal circumstances, free radicals attacking normal cells activates the secretion of antioxidant enzymes and still maintains redox balance. (**B**) In the presence of internal and external inducements, more free radicals and cytokines appear. The oxidative balance in the body cannot be maintained, forming an oxidative stress state, and inflammation may occur. There is always an oxidative stress state in the body, and the possibility of chronic inflammation gradually developing into cancer is greatly increased.

**Figure 2 pharmaceutics-14-01993-f002:**
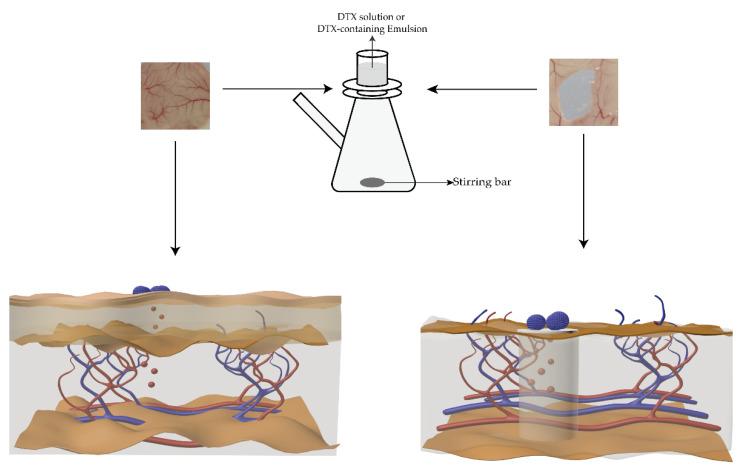
Schematic diagram of drug release experiments ex vivo. The left figure shows the DTX release from the emulsion across the mouse skin, and the right figure shows the DTX release from the emulsion across the mouse subcutaneous mucosa.

**Figure 3 pharmaceutics-14-01993-f003:**
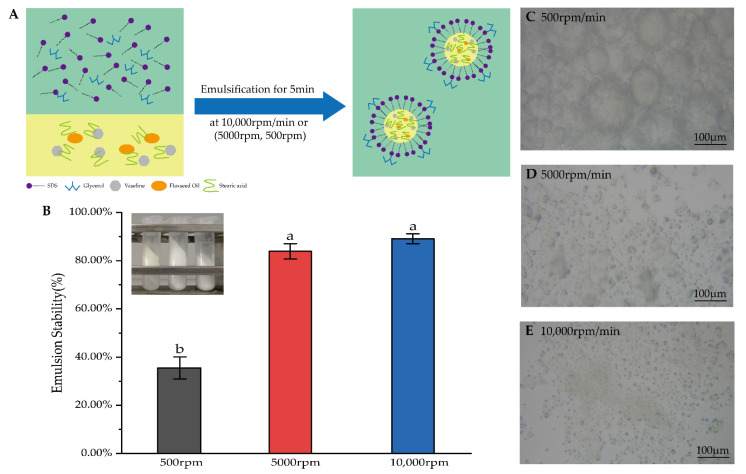
(**A**) Diagram of the preparation of the emulsion. (**B**) Results of accelerated emulsion stability and experimental figure at shear speeds of 500 rpm, 5000 rpm and 10,000 rpm. The data are means ± SD (n = 3). Means with different letters (a,b) are significantly different (*p* < 0.05) via Duncan’s test. (**C**–**E**) Microphotographs of emulsions were mixed at 500 rpm, 5000 rpm and 10,000 rpm.

**Figure 4 pharmaceutics-14-01993-f004:**
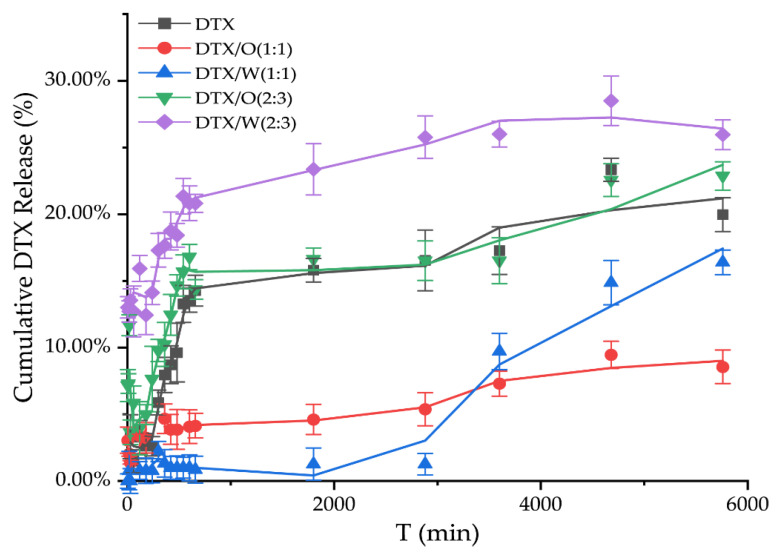
In vitro release studies of DTX solution and DTX-containing emulsions were based on the differences in the dosages of stearic acid and Vaseline in the emulsion, 2.5 g and 2.5 g (1:1), and the dosage was 1.0 g and 1.5 g (2:3) in the 10 mL system, respectively.

**Figure 5 pharmaceutics-14-01993-f005:**
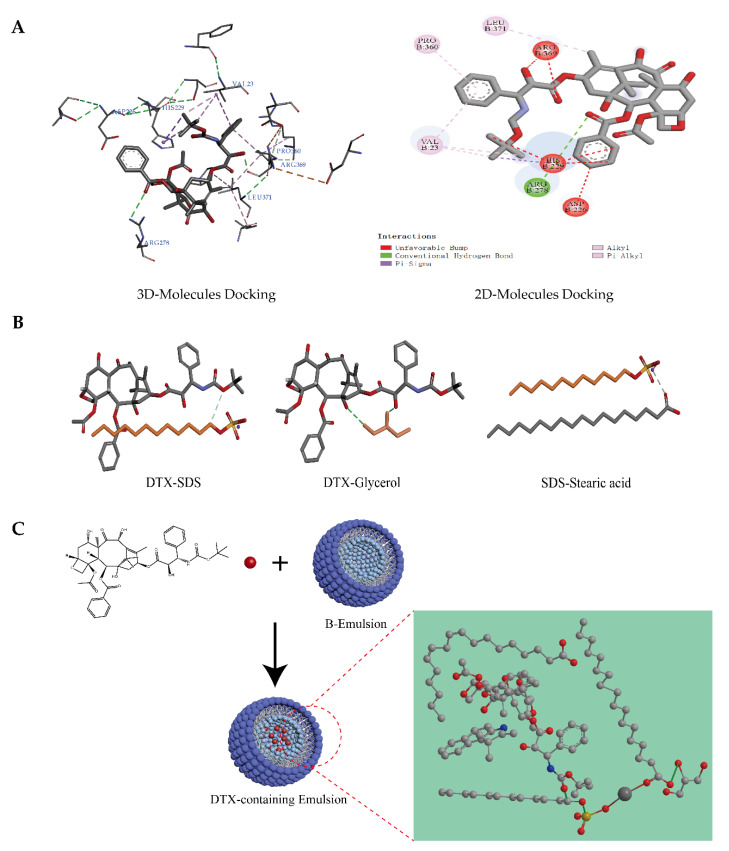
Prediction results by Discovery Studio 2016 software. (**A**) Three-dimensional and two-dimensional docking analysis binding β-tubulin (ID: 1TUB) with DTX. (**B**) Intermolecular docking analysis: DTX-Glycerol (hydrogen bond); DTX-SDS (hydrogen bond); SDS-Stearic acid (ionic bond). (**C**) Self-assembly layout and form of the micellar microparticles of DTX-containing emulsions.

**Figure 6 pharmaceutics-14-01993-f006:**
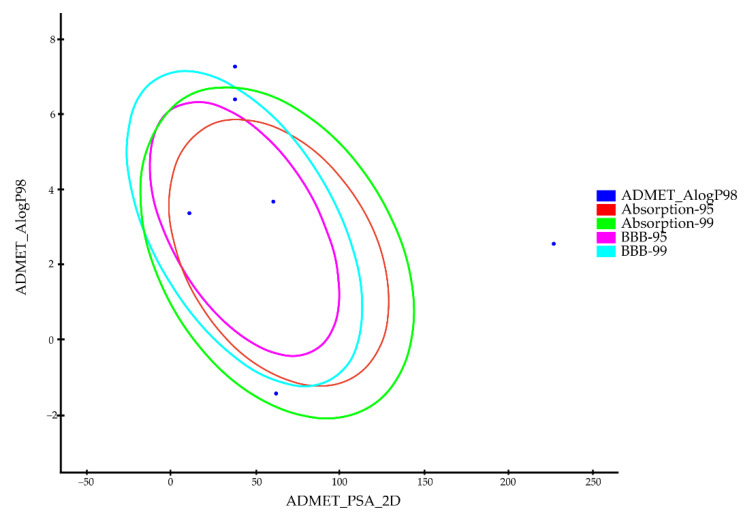
The plot of the two-dimensional polar surface area (PSA_2D) vs. the calculated ALogP98 for tested compounds showing the 95% and 99% confidence ellipses corresponding to the blood–brain barrier (BBB) and the human intestinal absorption (HIA) models.

**Figure 7 pharmaceutics-14-01993-f007:**
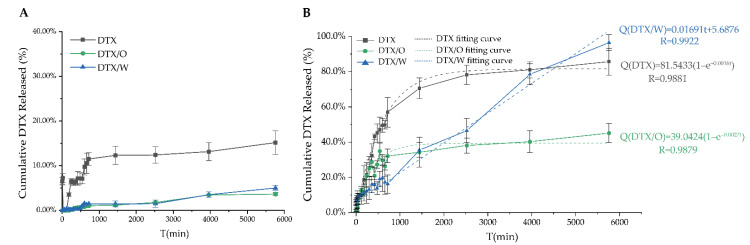
The transdermal release profiles of DTX solution, DTX/O-containing emulsion and DTX/W-containing emulsion in phosphate buffer saline containing 10% ethanol: (**A**) Mouse skin; (**B**) Mouse subcutaneous mucosa.

**Figure 8 pharmaceutics-14-01993-f008:**
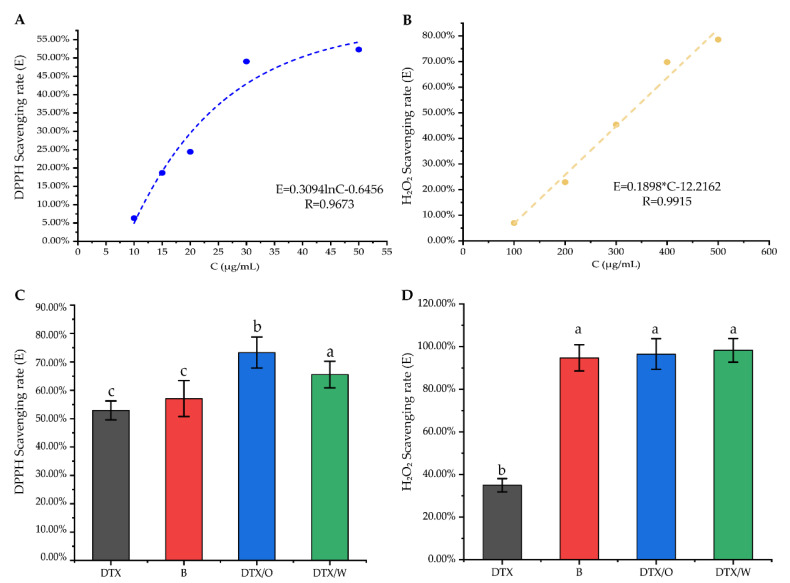
Antioxidant activity studies in vitro: (**A,B**) The relationship between DTX solution and DPPH scavenging rate (E%, blue) or H_2_O_2_ scavenging rate (E%, yellow); (**C**,**D**) DTX solution (DTX), blank emulsion (B), DTX/O-containing emulsion (DTX/O) and DTX/W-containing emulsion (DTX/W) had different scavenging rates on DPPH free radicals (DTX-containing concentration: 50 μg/mL) and H_2_O_2_ (DTX-containing concentration: 250 μg/mL). The data are means ± SD (n = 3). Means with different letters (a–c) are significantly different (*p* < 0.05) via Duncan’s test.

**Table 1 pharmaceutics-14-01993-t001:** Optimization of oil phase composition of DTX-containing emulsion via drug release in vitro.

Formulation	DTX (mL)	Oil Phase			Water Phase		
		Stearic Acid (g)	Vaseline (g)	Flaxseed Oil (g)	SDS (g)	Glycerol (g)	PBS
DTX (1:1)	1.0	2.5	2.5	-	0.1	1.5	to 10 mL
DTX (2:3)	1.0	1.0	1.5	0.3	0.1	1.5	

1:1 and 2:3 mean the dosage ratios of stearic acid and Vaseline in the formulation.

**Table 2 pharmaceutics-14-01993-t002:** ADME results of docetaxel and excipients.

Component	BBB Level ^a^	Solubility Level ^b^	Absorption Level ^c^	CYP2D6 Prediction ^d^	PPB Prediction ^e^
Docetaxel	4	2	3	NIN	√
Stearic acid	4	2	0	NIN	√
Vaseline	0	2	0	NIN	√
Flaxseed oil	0	2	0	NIN	√
SDS	1	3	0	NIN	√
Glycerol	4	4	1	NIN	×

^a^ BBB level, 0 = very high, 1 = high, 2 = medium, 3 = low and 4 = very low. ^b^ Solubility level, 1 = very low, 2 = low, 3 = good and 4 = optimal. ^c^ Absorption level, 0 = good, 1 = moderate, 2 = poor and 3 = very poor. ^d^ CYP2D6 is the cytochrome P2D6. The compound may be a CYP2D6 inhibitor (IN) or non-inhibitor (NIN). ^e^ PPB is the plasma protein binding that may be less than 90% (×) or more than 90% (√).

**Table 3 pharmaceutics-14-01993-t003:** Prediction of skin sensitivity and skin irritation in excipients.

Component	Skin Sensitization	Skin Irritancy
Stearic acid	None	Moderate
Vaseline	Strong	Mild
Flaxseed oil	None	Moderate
Glycerol	None	Mild
SDS	None	Mild

**Table 4 pharmaceutics-14-01993-t004:** Correlation coefficient values (R) for each mathematical model of DTX, DTX/O and DTX/W.

Group	Zero-Order	First-Order	Higuchi	Hixson–Crowell	Koresmeyer–Peppas
DTX	Q = 0.0235 t + 11.427 R = 0.7665	Q = 81.5433(1 − e^−0.0016 t^) R = 0.9881	Q = 1.5584 t^1/2^ + 2.8141 R = 0.9569	Q = 100 [1 − (1 − 0.00029 t)^3^] R = 0.9096	Q = 0.3045 t^0.4132^ R = 0.9689
DTX/O	Q = 0.0126 t + 4.073 R = 0.6611	Q = 39.0424(1 − e^−0.0027 t^) R = 0.9879	Q = 0.9384 t^1/2^ − 0.7333 R = 0.9181	Q = 100 [1 − (1 − 0.00013 t)^3^] R = 0.6984	Q = 1.1403 t^0.4642^ R = 0.9238
DTX/W	Q = 0.0169 t + 5.688 R = 0.9922	Q = 148.7101(1 − e^−0.0002 t^) R = 0.9276	Q = 1.001 t^1/2^ + 2.5097 R = 0.9504	Q = 100 [1 − (1 − 0.00011 t)^3^] R = 0.6729	Q = 1.9069 t^0.4262^ R = 0.9275

DTX means DTX solution. DTX/O means the emulsion containing DTX with oil phase. DTX/W means the emulsion containing DTX with water phase.

**Table 5 pharmaceutics-14-01993-t005:** Inhibition rates of MDA production by DTX and DTX/W-containing emulsions in different tissue homogenates.

Group	Inhibition Rate	
	Liver	Spleen
50 μg/mL DTX	-	43.90 ± 2.38% ^b^
100 μg/mL DTX	7.69 ± 0.72% ^b^	52.79 ± 1.80% ^ab^
200 μg/mL DTX	21.98 ± 1.18% ^a^	48.33 ± 1.72% ^ab^
500 μg/mL DTX	20.11 ± 1.13% ^a^	57.0.2 ± 1.51% ^a^
1000 μg/mL DTX	26.77 ± 1.38% ^a^	50.58 ± 1.11% ^ab^
DTX/W	26.21 ± 4.21% ^a^	56.59 ± 6.71% ^a^

The concentration of DTX/W-containing emulsion is 500 μg/mL. The data are means (*n* = 3) ± SD. Means with different letters (a,b) are significantly different (*p* < 0.05) via Duncan’s test.

## Data Availability

Not applicable.

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
