# Peer review of "The Fabrication of Docetaxel-Containing Emulsion for Drug Release Kinetics and Lipid Peroxidation"

_pharmaceutics, 2022, doi:10.3390/pharmaceutics14101993_

Round 1

Reviewer 1 Report

17.08.2022.

A review to evaluate its suitability for publication Type of manuscript:

Article
Title: Fabrication of Docetaxel-containing Emulsion for Local Skin Application: Drug Release Kinetics and Lipid Peroxidation Study

Authors: Yifang Wu , Mengmeng Wang , Yufan Li , Hongmei Xia , Yongfeng Cheng , Chang Liu , Ying Xia , Yu Wang , Yan Yue , Xiaoman Cheng , Zili Xie

The manuscript by Yifang Wu and co-workers described the Docetaxel (DTX)-based formulation.

The manuscript submitted for review is an innovative work devoted to the study of a new action mechanism of Docetaxel, which has been used in medicine as an antitumor agent for more than 30 years. The importance of this study lies in the development of new medicinal microsomal forms of API, the study of their stability and properties of ADME.

There are several questions, the answers to which will significantly improve the quality of the submitted manuscript material:

1.      The main question that caused the analysis of this manuscript is to substantiate the implementation of the antioxidant mechanism of antitumor docetaxel, as soon as possible, in the literature [1. Yang Z, Fong DW, Yin L, Wong Y, Huang W. Liposomes modulate docetaxel-induced lipid oxidation and membrane damage in human hepatoma cells. J Liposome Res. 2009;19(2):122-30. 2. Ray, S., Mondial, S., Ray, S.D. et al. Role of antioxidants on docetaxel-induced in vitra lipid peroxidation using malondialdehyde asmodel marker: an experimental and in silico approach. Med Chem Res 23, 4436–4446 (2014)] the opposite properties of cytostatics of taxa as derivatives of vincaalkaloids are described: docetaxel-induced lipid oxidation and membrane damage in human hepatoma cells.

2.      So, what was the basis for the authors to study the antioxidant properties of docetaxel? The answer can be given by analyzing the chemical structure of the taxane molecule, however, the authors do not give it.

3.      Line 74: DTX has one more phenolic hydroxyl group than PTX.

The authors point to the relationship between phenolic hydroxyls and possible antioxidant activity, probably similar to flavones?

On the docetaxel structure I have given, I kindly ask you to indicate the presence of a phenolic group:

Docetaxel https://chem.nlm.nih.gov/chemidplus/rn/114977-28-5

Benzenepropanoic acid, beta-(((1,1-dimethylethoxy)carbonyl)amino)-alpha-hydroxy-, 12b-(acetyloxy)-12-(benzoyloxy)-2a,3,4,4a,5,6,9,10,11,12,12a,12b-dodecahydro-4,6,11-trihydroxy-4a,8,13,13-tetramethyl-5-oxo-7,11-methano-1H-cyclodeca(3,4)benz(1,2-b)oxet-9-yl ester

4.      Line 75: However, there are few reports of the antioxidant activity of DTX.

It is necessary to provide links to these publications.

5.      The question arises about the ratio of therapeutic efficacy and toxicity of the developed forms of docetaxel in the treatment of various diseases, for example, psoriasis. Since, the literature [FarhaNG, Kasi A. Docetaxel. 2022 Feb 24] there is evidence of skin toxicity when using docetaxel: acral erythema, Peripheral neuropathy.

6.       Line 359: Figure 6. In vitro antioxidant activity studies (A, B)

What is the significance of the segments cut off by a straight line on the axes OY, OX in Fig. 6?

Respectfully, reviewer

Author Response

Response to Reviewer’s  Comments

Dear Reviewer:

     On behalf of my co-authors, we sincerely thank you for giving us an opportunity to revise the manuscript. We appreciate your valuable feedback that we have used to improve the quality of our manuscript entitled “Fabrication of Docetaxel-containing Emulsion for Local Skin Application: Drug Release Kinetics and Lipid Peroxidation Study”. We have studied comments carefully and have made corrections which we hope meet with approval. Revised portions are marked with red on the paper. The main corrections in the paper and the responses to your comments are as follows:

Point 1: The main question that caused the analysis of this manuscript is to substantiate the implementation of the antioxidant mechanism of antitumor docetaxel, as soon as possible, in the literature [1. Yang Z, Fong DW, Yin L, Wong Y, Huang W. Liposomes modulate docetaxel-induced lipid oxidation and membrane damage in human hepatoma cells. J Liposome Res. 2009;19(2):122-30. 2. Ray, S., Mondial, S., Ray, S.D. et al. Role of antioxidants on docetaxel-induced in vitra lipid peroxidation using malondialdehyde as model marker: an experimental and in silico approach. Med Chem Res 23, 4436–4446 (2014)] the opposite properties of cytostatics of taxa as derivatives of vinca alkaloids are described: docetaxel-induced lipid oxidation and membrane damage in human hepatoma cells.

Response 1: We sincerely appreciate the reviewer’s constructive suggestion. Disease development ranges from oxidative stress to inflammation, then to tumors. Docetaxel has been found to have great potential therapeutic effects in cancer treatment, and many studies on anti-tumor are profound. But tumor has been the worst result of disease development. Presently, the guideline for the administration of docetaxel in cancer chemotherapy is that 75 mg/cm2 of docetaxel is administered every three weeks, which can act on the whole body through intravenous injection and also cause the premise of systemic side effects.

The aim of our study is to treat some refractory skin diseases by low concentration local administration. Although the low concentration of DTX solution (50-500μg/mL) has played an inhibitory role in both in vitro free radical scavenging experiments and lipid peroxidation models, the increasing concentration can’t significantly inhibit the production of MDA. In addition, the in vitro and ex vitro release experiments showed that the DTX-containing emulsion was slow-release, reducing the damage to surrounding normal cells.

Our research idea is to focus on the disease from the early stage of disease development, which can prevent the disease from developing to the tumor stage earlier. And we have added Figure 1 in the manuscript to describe this research point which is more novelty and meaningful.

Point 2: So, what was the basis for the authors to study the antioxidant properties of docetaxel? The answer can be given by analyzing the chemical structure of the taxane molecule, however, the authors do not give it.

Response 2: We are grateful to the reviewer for pointing out this problem. We analyzed the molecular structure of taxane molecules to give the basis for studying the antioxidant properties of docetaxel. The substituents at the C10 position of the two are different, paclitaxel is acetyl, and docetaxel is hydroxyl (lines 79-83)

Point 3: Line 74: DTX has one more phenolic hydroxyl group than PTX.

The authors point to the relationship between phenolic hydroxyls and possible antioxidant activity, probably similar to flavones?

On the docetaxel structure I have given, I kindly ask you to indicate the presence of a phenolic group:

Docetaxel https://chem.nlm.nih.gov/chemidplus/rn/114977-28-5

Benzenepropanoic acid, beta-(((1,1-dimethylethoxy) carbonyl) amino)-alpha-hydroxy-, 12b-(acetyloxy)-12-(benzoyloxy)-2a,3,4,4a,5,6,9,10,11,12,12a,12b-dodecahydro-4,6,11-trihydroxy-4a,8,13,13-tetramethyl-5-oxo-7,11-methano-1H-cyclodeca (3,4) benz(1,2-b) oxet-9-yl ester

Response 3: Thank you for your careful suggestion. We are sorry that there is an expression error in this sentence “DTX has one more phenolic hydroxyl group than PTX”. The sentence has been revised to “From the perspective of chemical structure, based on the mother nucleus of taxane, the substituents at the C10 position of the two are different, paclitaxel is acetyl, and docetaxel is hydroxyl.” (lines 79-81)

According to the chemical structure given by the reviewer, the hydroxyl group is located at the C6 position. Compared with the stable free radical structure of the hydroxyl group, the acetyl group has stronger electron adsorption, which may reduce the antioxidant activity. Therefore, we speculate that docetaxel may have antioxidant capacity.

Point 4: Line 75: However, there are few reports of the antioxidant activity of DTX.

It is necessary to provide links to these publications.

Response 4: Thanks for your careful review. We found the literature [Zhang, Y.; Fang, Y.; Cheng, Z. et al. Free Radical Scavenging Activities of the Extracts from Taxus Chinensis var. mairei. Asian Journal of Chemistry 2013, 25, 6213-6215] that reported the antioxidant activity of paclitaxel. Due to docetaxel being the second-generation taxane, we speculate that docetaxel has antioxidant potential. The experimental study showed that the IC50 of DPPH radical and hydrogen peroxide were 40.65 ± 6.29μg/mL and 327.47 ± 52.23μg/mL. This result is superior to the half clearance rate of paclitaxel reported in the literature.

Point 5: The question arises about the ratio of therapeutic efficacy and toxicity of the developed forms of docetaxel in the treatment of various diseases, for example, psoriasis. Since, the literature [FarhaNG, Kasi A. Docetaxel. 2022 Feb 24] there is evidence of skin toxicity when using docetaxel: acral erythema, Peripheral neuropathy.

Response 5: Thanks for your suggestion. We cannot deny that docetaxel has a series of side effects in cancer chemotherapy, including skin toxicity. It is the problem that all researchers want to solve and prevent. In the literature [FarhaNG, Kasi A. Docetaxel. 2022 Feb 24] there described that the dose for indications such as breast cancer, non-small cell lung cancer, prostate cancer, and advanced gastric cancer is 75mg/m2. The normal body surface area of an adult is 1.5-1.7 square meters, which is equivalent to 112.5-127.5mg for a single dose. However, the emulsion we studied had a drug content of 5mg/mL. The possibility of skin toxicity can be reduced by local administration of a small dose and slow release of emulsion. The follow-up study will focus on local administration to treat refractory skin diseases, such as psoriasis.

Point 6: Line 359: Figure 6. In vitro antioxidant activity studies (A, B)

What is the significance of the segments cut off by a straight line on the axes OY, OX in Fig. 6?

Response 6: Thanks for your suggestion. Because a picture was added in the introduction section, the original Figure 6 was changed to Figure 7 in the manuscript. In Figure 7. A and B, the segments cut off by a straight line on the axes OY and OX represent the relationship between DTX concentration (C) and scavenging rate (E). The relationship between DTX concentration and DPPH scavenging rate in Figure 7A was logarithmic. In Figure 7B, there was a linear relationship between DTX concentration and hydrogen peroxide clearance. Moreover, these showed that the clearance rate was affected by DTX concentration.

Special thanks to you for your good comments.

Reviewer 2 Report

The work appears interesting even if some parts need to be deepened. Particle systems such as nanomicles have been widely exploited in recent years to increase the solubility of insoluble drugs also with antitumor activity and to reduce their toxicity. The authors can insert the following 2 citations (DOI: 10.3390/molecules24091793; 10.3390/pharmaceutics12111078) in the introduction.

Regarding the use of animals, authors should report the authorization number of the document authorizing them for use.

Please describe in more detail the equipment used for in vitro release and skin permeation studies. Better specify which part of the animal's body is used for permeation studies.

Describe in detail the characteristics of the subcutaneous mucosa.

How is the drug analyzed? Report in detail the analytical method.

In the results paragraph, the results obtained are merely described but there is no discussion and no hypothesis is made on why these results are obtained with reference to the data present in the literature.

Author Response

Response to Reviewer’s Comments

Dear Reviewer:

     On behalf of my co-authors, we sincerely thank you for giving us an opportunity to revise the manuscript. We appreciate your valuable feedback that we have used to improve the quality of our manuscript entitled “Fabrication of Docetaxel-containing Emulsion for Local Skin Application: Drug Release Kinetics and Lipid Peroxidation Study”. We have studied comments carefully and have made corrections which we hope meet with approval. Revised portions are marked with red on the paper. The main corrections in the paper and the responses to your comments are as follows:

Point 1: The work appears interesting even if some parts need to be deepened. Particle systems such as nanomicles have been widely exploited in recent years to increase the solubility of insoluble drugs also with antitumor activity and to reduce their toxicity. The authors can insert the following 2 citations (DOI: 10.3390/molecules24091793; 10.3390/pharmaceutics12111078) in the introduction.

Response 1: We sincerely appreciate the reviewer’s careful suggestion. We have supplemented the description (lines 53-55) and cited these two citations in the introduction [20,21].

Point 2: Regarding the use of animals, authors should report the authorization number of the document authorizing them for use.

Response 2: Thanks for the reviewer’s suggestion. The authorization number of the animals’ document is SYXK(wan)2020-001.

Point 3: Please describe in more detail the equipment used for in vitro release and skin permeation studies. Better specify which part of the animal's body is used for permeation studies.

Response 3: Thanks for the reviewer’s careful suggestion. We have carefully and completely described the equipment used for in vitro release and skin permeation studies (lines 153-166, 211-215).

         Mouse dorsal subcutaneous mucosa was used for permeation studies. This experiment is based on the literature [XIA H, CHENG Z, CHENG Y, et al. Investigating the passage of tetramethylpyrazine-loaded liposomes across blood-brain barrier models in vitro and ex vivo [J]. Mater Sci Eng C Mater Biol Appl, 2016, 69(1010-7)]

Point 4: Describe in detail the characteristics of the subcutaneous mucosa.

Response 4: Thanks for your kind review. Compared with the stratum corneum, the subcutaneous mucosa has great permeability. In addition, there are many places on the body where the drug can be applied externally to the mucosa, such as the eye mucosa and the nasal mucosa. This is also a mode of administration that we want to continue to study in addition to skin administration.

Point 5: How is the drug analyzed? Report in detail the analytical methodipheral neuropathy.

Response 5: Thanks for the reviewer’s suggestion. In the experiment, the standard curve of docetaxel was prepared by ultraviolet spectrophotometry. The concentration of docetaxel was converted by a standard curve and the drug content was analyzed. In terms of computers, ADME and toxicity protocol in Discovery Studio 2016 are used to predict the absorption, distribution, metabolism, excretion, skin toxicity, and skin irritation of drugs.

Peripheral neuropathy is a common side effect of docetaxel in chemotherapy. In the experiment, we will pay attention to the prescription research in the early stage, and in the later stage, we will focus on the study of docetaxel emulsion on skin compliance. In the study of paclitaxel ointment in the treatment of psoriasis, Yin et al reported that the skin of mice did not show any signs of skin toxicity or irritation after 12 days of administration. Considering the 3R principle of animal ethics, when we did not determine the treatment method for skin diseases, we used computer software to simulate and predict and didn’t conduct experiments on animals.

Point 6: In the results paragraph, the results obtained are merely described but there is no discussion and no hypothesis is made on why these results are obtained with reference to the data present in the literature.

Response 6: Thanks for the reviewer’s constructive suggestion. The experimental results are discussed in detail and some references are given.

Special thanks to you for your good comments.

Reviewer 3 Report

The current manuscript title “Fabrication of Docetaxel-containing Emulsion for Local Skin 2 Application: Drug Release Kinetics and Lipid Peroxidation 3 Study” focus on transdermal delivery of DTX. This manuscript needs major revision before publication.

1.       Add more data into abstract section.

2.       Why Kunming mice were used in this study?? Any specific reason?? Please mention in methodology.

3.       What is the %EE of prepared emulsion???

4.       How microphotographs of emulsion were taken?? Please add the method with reference if any??

5.       What was the logic behind the formulation of blank emulsion??

6.       What is pH of PBS solution used for in vitro release study??? Why temperature 37 degree was used?? whereas the temperature of skin surface is 32 degree.

7.       What is pH of PBS solution containing 10% Ethanol for ex vivo study???

8.       What technique/chemical was used to enhance the permeability of drug across the skin??

9.       Add reference to the toxicity study section 2.7.

1.   For release kinetics, different models were used. why?? Whynot Koresmeyer-Peppas model which gives mechanism of release of drug from the dosage form??

1.   The discussion part needs more references and compare your results with literature.

1.   Conclusion is too lengthy, please summarize.

Author Response

Response to Reviewer’s Comments

Dear Reviewer:

     On behalf of my co-authors, we sincerely thank you for giving us an opportunity to revise the manuscript. We appreciate your valuable feedback that we have used to improve the quality of our manuscript entitled “Fabrication of Docetaxel-containing Emulsion for Local Skin Application: Drug Release Kinetics and Lipid Peroxidation Study”. We have studied comments carefully and have made corrections which we hope meet with approval. Revised portions are marked with red on the paper. The main corrections in the paper and the responses to your comments are as follows:

Point 1: Add more data into abstract section.

Response 1: We sincerely appreciate the reviewer’s suggestion. We have revised the abstract carefully and elaborated our work more completely.

Abastract: Docetaxel (DTX)-based formulation development is still confronted with significant challenges, due to its refractory solubility and side effects on normal tissues. Inspired by the application of the transdermal drug delivery model to topical treatment, we developed a biocompatible and slow-release DTX-containing emulsion via self-assembly prepared by a high-speed electric stir-ring method and optimized the formulation. The results of accelerated emulsion stability experiment showed that the emulsion prepared at 10000rpm/min had a stability of 89.15 ± 2.05%. The ADME, skin irritation, skin toxicity, and molecular interaction between DTX and excipients were predicted via Discovery Studio 2016 software. In addition, DTX addition in different phases of the emulsion showed differences in ex vivo release. The DTX release ex vivo of DTX/O-containing emulsion and DTX/W-containing emulsion were 45.07 ± 5.41% and 96.48 ± 4.54%, respectively. In vitro antioxidant assays and anti-lipid peroxidation models have revealed the antioxidant potential of DTX. However, DTX-containing emulsions could maintain and even enhance the antioxidant effect, both it scavenged free radicals in vitro and inhibited the process of lipid peroxidation.

Point 2: Why Kunming mice were used in this study?? Any specific reason?? Please mention in methodology.

Response 2: Thanks for the reviewer’s suggestion. Kunming mice are widely used as experimental animals in China. It has a large gene pool, high gene heterozygosity rate, low tumor spontaneous rate, strong disease resistance and adaptability, and high reproduction and survival rate. In this study, we took mouse dorsal skin, spleen, and liver for ex vivo experiments instead of in vivo modeling, so we chose the Kunming mice commonly used in the laboratory.

Point 3: What is the %EE of prepared emulsion???

Response 3: Thanks for the reviewer’s suggestion. We have supplemented the experiment method and results of the EE of prepared emulsion in section 2.3.3 and section 3.2.

Point 4: How microphotographs of emulsion were taken?? Please add the method with reference if any??

Response 4: Thanks for the reviewer’s kind suggestion. We have supplemented the description of the method according to the study by Lee et al. in section 2.3.2 (lines 132-134).

Point 5: What was the logic behind the formulation of blank emulsion??

Response 5: We thank the reviewer for bringing our attention to the logic behind the formulation of blank emulsion. Vaseline and glycerin are common ingredients in emulsions and play a moisturizing role. Sodium dodecyl sulfate acts as a surfactant to emulsify. Stearic acid is a macromolecular compound and plays an auxiliary role in emulsification. Finally, flaxseed oil has skin affinity and excellent antioxidant capacity. In addition, the literature reported that the DTX-containing flaxseed oil lipid emulsion has more drug carrying capacity, and reduces the possibility of side effects in anti-tumor research

Point 6: What is pH of PBS solution used for in vitro release study??? Why temperature 37 degree was used?? whereas the temperature of skin surface is 32 degree.

Response 6: Thanks for the reviewer’s suggestion. The pH of PBS solution used for in vitro release study was 7.4. The temperature of the receiving solution in the lower compartment of the diffusion cell was maintained at 37±1°C in order to simulate the drug release under the physiological environment.

In the literature: [CHENG M, LIU Q, GAN T, et al. Nanocrystal-Loaded Micelles for the Enhanced In Vivo Circulation of Docetaxel [J]. Molecules, 2021, 26(15); FOULADIAN P, AFINJUOMO F, ARAFAT M, et al. Influence of Polymer Composition on the Controlled Release of Docetaxel: A Comparison of Non-Degradable Polymer Films for Oesophageal Drug-Eluting Stents [J]. Pharmaceutics, 2020, 12(5); MAO K, ZHANG W, YU L, et al. Transferrin-Decorated Protein-Lipid Hybrid Nanoparticle Efficiently Delivers Cisplatin and Docetaxel for Targeted Lung Cancer Treatment [J]. Drug Des Devel Ther, 2021, 15(3475-86)] the in vitro release method described that the samples were soaked in PBS buffer solution (pH=7.4) and the experiment was carried out at 37 degrees.  In the skin penetration study, although we considered the skin temperature of mice, we chose to control the receiving solution at 37±1°C to ensure that it was close to the physiological environment. The temperature of the upper compartment was also maintained at 30±2℃ under the influence of the indoor environment.

 Point 7: What is pH of PBS solution containing 10% Ethanol for ex vivo study???

Response 7: Thanks for the reviewer’s suggestion. The pH of PBS solution containing 10% Ethanol was about 7.4. 10% ethanol was added into the diffusion medium of the lower compartment to form the sink conditions and induce DTX release.

Point 8: What technique/chemical was used to enhance the permeability of drug across the skin??

Response 8: Thanks for the reviewer’s suggestion. Sodium dodecyl sulfate (SDS) is a kind of anionic surfactant, which can promote permeability by changing the lipid bimolecular structure of the stratum corneum, removing keratin, and expanding the stratum corneum.

Point 9: Add reference to the toxicity study section 2.7.

Response 9: Thanks for your suggestion. We have added more references in the toxicity study section 2.7. These references are listed as follows:

  1. Singh, S.; Das, T.; Awasthi, M.; Pandey, V.P.; Pandey, B.; Dwivedi, U.N. DNA topoisomerase-directed anticancerous alkaloids: ADMET-based screening, molecular docking, and dynamics simulation. Biotechnol Appl Biochem 2016, 63, 125-137, doi:10.1002/bab.1346.
  2. Pires, D.E.; Blundell, T.L.; Ascher, D.B. pkCSM: Predicting Small-Molecule Pharmacokinetic and Toxicity Properties Using Graph-Based Signatures. J Med Chem 2015, 58, 4066-4072, doi:10.1021/acs.jmedchem.5b00104.

Point 10: For release kinetics, different models were used. why?? Why not Koresmeyer-Peppas model which gives mechanism of release of drug from the dosage form??

Response 10: Thanks for your suggestion. Because we prepared two kinds of emulsions and DTX was added to the oil phase and the water phase respectively, we used a variety of mathematical models to analyze the kinetics and mechanism of DTX release from the emulsion. We have supplemented the fitting results of the Koresmeyer-peppas model in the manuscript. However, according to the correlation coefficient values (R) of zero-order, first-order, Higuchi, Hixson-Crowell, and Koresmeyer-Peppas model, the release of DTX solution and DTX from DTX/O was close to the first-order release equation, and the release of DTX from DTX/W was close to the zero-order release equation.

Point 11: The discussion part needs more references and compare your results with literature.

Response 11: Thanks for your suggestion. The experimental results are discussed in detail and some references are given.

Point 12: Conclusion is too lengthy, please summarize.

Response 12: Thanks for your suggestion. We have revised carefully and summarized the conclusions as follows:

In summary, DTX was delivered with microemulsion as the carrier for local skin application. There is mainly van der Waals force existed between each component in the emulsion, and a small amount of hydrogen bonds improve the stability and permeability of the emulsion. Drug release across mouse subcutaneous mucosa ex vivo experiments showed that there was a difference between the addition of DTX to the oil phase or water phase. When DTX was in the oil phase, the release of DTX in the DTX/O was from the micelles. When DTX was in the water phase, the release amount of DTX in the DTX/W was twice that of the former, which is also close to the ideal drug release system. Furtherly, free radical scavenging experiments in vitro showed that both DTX and DTX-containg emulsion had antioxidant capacity, and there was a synergistic effect between them. Although increasing the concentration of DTX did not regularly control lipid peroxidation, DTX affected the production of oxidation products as a whole. Therefore, taking emulsion as the platform for DTX delivery, targeted treatment of some refractory chronic skin diseases, such as psoriasis, makes it more meaningful in clinical practice.

Special thanks to you for your good comments.

Round 2

Reviewer 2 Report

The authors must better justify the use of a mucosa as a substitute for a skin that possesses the stratum corneum and which represents the greatest barrier to permeation. In my opinion, the mucosa is not appropriate as skin mode, the authors have to report suitable references to support their choice.  The reference they reported concerns a study at the level of the brain (blood-brain barrier), so the choice of the mucosa is certainly more appropriate. The results may not be relevant once the formulation is applied to the skin.

Author Response

Response to Reviewer’s Comments

Dear Reviewer:

     We sincerely thank you for giving us an opportunity to revise the manuscript. We thank you for your valuable feedback and have carefully studied your comments. According to these comments, we revised the title “Fabrication of Docetaxel-containing Emulsion for Local Skin Application: Drug Release Kinetics and Lipid Peroxidation Study” to “Fabrication of Docetaxel-containing Emulsion for Drug Release Kinetics and Lipid Peroxidation Study”, made corrections to improve the quality of the manuscript which we hope to get approval. The modified part is marked in red on the paper. The main corrections in the document and the responses to your comments are as follows:

Point 1: The authors must better justify the use of mucosa as a substitute for a skin that possesses the stratum corneum and which represents the greatest barrier to permeation. In my opinion, the mucosa is not appropriate as skin mode, the authors have to report suitable references to support their choice.  The reference they reported concerns a study at the level of the brain (blood-brain barrier), so the choice of the mucosa is certainly more appropriate. The results may not be relevant once the formulation is applied to the skin.

Response 1: We appreciate your comments and agree that mucosa is unsuitable for skin modes. Here I would like to give a response to the transmembrane release experiment.

First of all, this manuscript focused on the preparation of docetaxel-containing emulsion and the exploration of its antioxidant potential. The application of skin administration mentioned in the title is an extension of the follow-up study of the emulsion, but it now seems inappropriate. Therefore, we changed the title to “Fabrication of Docetaxel-containing Emulsion for Drug Release Kinetics and Lipid Peroxidation Study”.

Secondly, we did transdermal release experiments. In the transdermal release experiment, DTX release from DTX solution was 15.31 ± 2.79%. While, only a small amount of docetaxel was detected in DTX/O-containing emulsion and DTX/W-containing emulsion within 96h, which was 3.97 ± 0.34% and 5.64 ± 0.22%, respectively. Then, using sonication to handle experimental skin, the amount of docetaxel retained in mouse skin was 72.57 ± 6.02% for DTX/O and 61.96 ± 5.11% for DTX/W, which indicated that the emulsion micelle structure made the slow release of DTX. In addition, the retention of inflammatory skin was higher than that of normal skin by Yin et al. reported, which also showed that the inflammatory environment could improve the permeation and induce DTX enrichment.

Then, we selected subcutaneous mucosa as a substitute for the penetration study to conduct penetration experiments to explore how the drug dose enters the systemic circulation. After all, in addition to the skin, the subcutaneous mucosa such as ocular mucosa, nasal mucosa, and oral cavity mucosa is the ideal route of administration. The drugs across the mucosa of ocular, nasal, and oral cavity reach the target site or directly enter the systemic circulation. The curve between time and cumulative drug content percentage was drawn and fitted according to various release models in the drug release experiment across subcutaneous mucosa. The results showed that the two emulsions had different release behaviors. The release behavior of DTX/O-containing emulsion was consistent with that of DTX solution, which conformed to the first-order release equation. DTX/W-containing emulsion conformed to the zero-order release equation. More the purpose of transdermal and transmucosal release experiments was to provide a new therapeutic idea for a certain disease and hope to lay a foundation for future research.

Finally, we have supplemented the methods of transdermal experiments in section 2.9 (lines 212-216) and schematic diagram of drug release experiments ex vivo (Figure 2). In addition, we have added the results and analysis of drug release ex vivo in section 3.5. (lines 352-356, 360-372)

Figure 2. Schematic diagram of drug release experiments ex vivo. The left figure showed the DTX release from the emulsion across the mouse skin, and the right figure showed the DTX release from the emulsion across the mouse subcutaneous mucosa.

We tried our best to improve the manuscript and made some changes in the manuscript. Once again thank you very much for your comments and suggestions.

Reviewer 3 Report

The authors have incorporated all the suggestions and manuscript is now improved.  Therefore I recommend this manuscript for publication. 

Regards 

Author Response

Response to Reviewer’s Comments

Dear Reviewer:

    We sincerely thank you for giving us an opportunity to revise the manuscript. We appreciate your valuable feedback that we have used to improve the quality of our manuscript entitled “Fabrication of Docetaxel-containing Emulsion for Drug Release Kinetics and Lipid Peroxidation Study”. We have studied comments carefully and have made corrections. The main corrections in the paper and the responses to your comments are as follows:

Point 1: The authors have incorporated all the suggestions and manuscript is now improved. Therefore, I recommend this manuscript for publication.

Response: We are very grateful for your review. My co-author and I sincerely thank you. We are very happy to receive your consent, and once again express our gratitude to you on behalf of all the co-authors.

Round 3

Reviewer 2 Report

accepted

Author Response

Response to Reviewer 2 Comments

Dear Reviewer:

    We sincerely thank you and appreciate your valuable feedback to our manuscript entitled “Fabrication of Docetaxel-containing Emulsion for Drug Release Kinetics and Lipid Peroxidation Study”.

Point 1: accepted.

Response 1: We are very grateful for your review. My co-author and I sincerely thank you. We are very happy to receive your consent, and once again express our gratitude to you on behalf of all the co-authors.